# A Spatial-Temporal Resolved Validation of Source Apportionment by Measurements of Ambient VOCs in Central China

**DOI:** 10.3390/ijerph17030791

**Published:** 2020-01-28

**Authors:** Longjiao Shen, Zuwu Wang, Hairong Cheng, Shengwen Liang, Ping Xiang, Ke Hu, Ting Yin, Jia Yu

**Affiliations:** 1School of Resource and Environmental Sciences, Wuhan University, Wuhan 430079, China; shenlj@whu.edu.cn (L.S.); chenghr@whu.edu.cn (H.C.); 2Environmental Monitoring Center of Wuhan, Wuhan 430022, China; when2003lsw@163.com (S.L.); hk0205@163.com (K.H.); tingyin_1405@163.com (T.Y.); yujia1253@126.com (J.Y.); 3Nanjing Intelligent Environmental Sci-Tech Company Limited, Nanjing 211800, China; xiangping@ies-tech.cn

**Keywords:** VOCs, source apportionment, PMF, Wuhan

## Abstract

Understanding the sources of volatile organic compounds (VOCs) is essential in the implementation of abatement measures of ground-level ozone and secondary organic aerosols. In this study, we conducted offline VOC measurements at residential, industrial, and background sites in Wuhan City from July 2016 to June 2017. Ambient samples were simultaneously collected at each site and were analyzed using a gas chromatography–mass spectrometry/flame ionization detection system. The highest mixing ratio of total VOCs was measured at the industrial site, followed by the residential, and background sites. Alkanes constituted the largest percentage (>35%) in the mixing ratios of quantified VOCs at the industrial and residential sites, followed by oxy-organics and alkenes (15–25%).The values of aromatics and halohydrocarbons were less than 15%. By contrast, the highest values of oxy-organics accounted for more than 30%. The model of positive matrix factorization was applied to identify the VOC sources and quantify the relative contributions of various sources. Gasoline-related emission (the combination of gasoline exhaust and gas vapor) was the most important VOC-source in the industrial and residential areas, with a relative contribution of 32.1% and 40.4%, respectively. Industrial process was the second most important source with a relative contribution ranging from 30.0% to 40.7%. The relative contribution of solvent usage was 6.5–22.3%. Meanwhile, the relative contribution of biogenic emission was only within the range of 2.0–5.0%. These findings implied the importance of controlling gasoline-related and industrial VOC emissions in reducing the VOC emissions in Wuhan.

## 1. Introduction

As major precursors of photochemical smog, volatile organic compounds (VOCs) are crucial in atmospheric chemistry. VOCs can react with NOx to form ozone, and take part in the formation of secondary organic aerosols (SOAs) through photochemical reactions and gas-phase particle reactions [1,2]. Research on pollution characteristics and VOC sources can provide insights into the pollution control of urban photochemicals and fine particulate matter [3,4].

However, the sources of VOCs are often complex because of various species, polluting industries and emission, sources largely depending on the levels of local energy consumption levels and the industrial structures [5,6]. Automobile exhaust, solvent usage, fuel evaporation, technological process in industries, and biomass burning are the major sources of anthropogenic regional emission [7,8,9]. Natural emission is also important in VOC sources. Thus, the estimation of VOC emission remains highly uncertain [10,11]. Positive matrix factorization (PMF) analysis is an effective method for identifying VOC sources and quantifying the relative contributions of various sources. Extensive research on resolving VOC sources through PMF have been conducted in recent years in different regions of China, including Beijing–Tianjin–Hebei, the Yangtze River Delta, and the Pearl River Delta [5,12,13,14,15,16,17]. However, limited studies are available on VOC resources in central China [18,19]. As the capital city of Hubei Province in central China, Wuhan has more than 10 million residents and is characterized by its unique terrain and a rapidly growing economy. In recent years, the number of registered motor vehicles have rapidly increased from ~1300 thousand in 2012 to >3200 thousand in 2019. With the rapid development of technological industries and the number of automobiles, Wuhan is facing severe haze pollution [20]. Petrochemical industries, such as basic chemicals, optoelectronics, car spraying, and printing, worsen the situation. Thus, Wuhan has been chosen as the study region for investigating the mechanism and related scientific issues of VOCs. However, previous studies on the source characteristics of VOCs have been conducted in residential areas [21]. Sources are accurately resolved by using the measurements of different functional areas due to the high complexity of emission sources. Therefore, this study mainly aims to acquire the characteristics and sources of VOCs from three different functional areas of Wuhan from July 2016 to June 2017, and the source categories of VOCs will be identified through the PMF model. Understanding the mixing ratios of VOC species will provide support to the local government in taking effective measures for the reduction of VOCs and O3 not only in Wuhan but also in other highly polluted regions.

## 2. Materials and Methods 

### 2.1. Information about the Sample Site

Wuhan has 10 national environmental automatic air quality monitoring stations. The VOC samples were collected by three national control monitoring stations (Figure 1).These monitoring stations were grouped into three categories on the basis of their geographic locations corresponding to the (1) residential site (shown as the Zi-yang [ZY] site and located in the central area of Wuhan; 114°18″ E, 30°30″ N), (2) industrial site (shown as the Zhuan-kou [ZK] site and located in the Wuhan Economic and Technological Development Area; 114°9″E, 30°28″N), and (3) background site (shown as the Mulan Lake [ML] site and located in the northwestern suburb area of the city, which was approximately 50 km from central Wuhan; 114°24″ E, 316″ N). The three types of sites reflected the VOC characteristics from different sources.

Environmental instantaneous samples were collected for three days each month from July 2016 to June 2017 at 9:00 a.m. and 15:00 p.m. local time. The specific collection time of each month’s environmental samples follows two main principles: On the one hand, the sampling date should-cover every month in the entire year, including both working days and weekends. The system sampling design was adopted with minimal human intervention to ensure the time representativeness of the collected samples. On the other hand, sampling in extreme weather such as wind, rainfall, and snowfall, should be avoided to rule out the accidental effects-caused by such extreme conditions. The meteorological parameters during the sampling periods were summarized in Table 1. A total of 359 valid VOC data were obtained from the regional measurements, with the exclusion of abnormal samples.

### 2.2. Instrument and Methods

Ambient VOC samples were instantaneously collected using 3.2 L fused silica stainless steel canisters, which had been precleaned with high-purity nitrogen (purity N 99.999%) and evacuated with an automated canister cleaner. A flow-limiting valve was used to collect instantaneous samples. A total of 101 VOCs species were analyzed by a three-stage cryofocusing preconcentration system (Entech 7200; Entech Instruments Inc., Simi Valley, CA, USA) coupled with a gas chromatography–mass spectrometry/flame ionization detection (GC–MS/FID) system (TH-300B, Wuhan Tianhong Instrument Co., Ltd., Hubei, China), composed of 28 alkanes, 11 alkenes, onealkyne, 17 aromatic compounds, nineoxy-organics, 34 halohydrocarbons and carbon disulfide. First, the samples were pumped into apreconcentration system in two ways. A Teflon filter was-utilized to prevent particulate matters from entering the system, and a water trap and an carbon dioxide removal tube were used to remove H_2_O and CO_2_. Second, the cryofocus unit was cooled down to −160 °C with liquid nitrogen to trap the VOCs in the air samples. Therefore, the VOC components were trapped-in the GC system. The system was equipped with two columns and two detectors, in which the C2–C5 non-methane hydrocarbons(NMHCs) were separated with a PLOT(Al_2_O_3_/KCl, 15 m × 0.32 mm ID) column and quantified by FID, whereas the C5–C12 hydrocarbons were separated via a semipolar column (DB-624, 30 m × 0.25 mm ID, J&W Scientific (Palo Alto, CA, USA)) and quantified by using aquadrupole MS detector. The entire process took ~41.6 min. Pure helium (purity N 99.999%) was used as the carrier gas. MSD was operated in the scan mode with amass range of 29–350 amu. The ionization method was applied to determine the electron impact (EI, 70 eV). The standard gas including the Photochemical Assessment Monitoring Stations (PAMS)—(57 NMHCs) and TO-15 s—(65 compounds), standard mixtures, purchased from Linde Spectra Environment Gases (Danbury, CT, USA) is used to calibrate the C2–C12 VOCs. Calibration was performed at five different concentrations from 0.5 to 8 ppbv by the 57 PAMS gas standard and 65 TO-15 gas standard. Bromochloromethane,1, 4-difluorobenzene, chlorobenzene-d5 and 4-bromofluorobenzene were used as the internal standards for each sample to calibrate the system. The precision of each species was within 10%. A gas standard (diluting from 1 ppm to 2 ppbv) was measured daily to check the stability of the system. The method detection limits-of various compounds ranged from 2 to 50 pptv.

### 2.3. PMF Model

PMF (V5.0, US EPA, Washington, DC, USA) is a widely used receptor model for the source apportionment of air pollution. The PMF model decomposes a matrix of element data into two matrices using Equation (1), wherein the actor contributions and factor profiles that must-be interpreted by an analyst into specific source types are represented using the measured source profile information, wind direction analysis, and emission inventories. This method is briefly introduced here and more detail will be found elsewhere [22,23]:(1)$$X_{ij}=\sum_{p=1}^{k}g_{ik}f_{ki}+e_{ij}$$
where, $x_{ij}$ is a data matrix *X* of i by j dimensions, in which i number of samples and j chemical species were measured. p is the number of source, f and g are the species profile of each source and the amount of mass contributed by each source to the sample $e_{ij}$ is the residual for each sample and specie [22].

The other details of PMF applied to VOC data for source profiles and the contributions of each VOC species have been introduced in previous studies [24,25,26]. In this study, 28 VOC species were input into the PMF model. The equation-based uncertainty file provides species-specific parameters to calculate uncertainties for each sample. Values below the method detection limit (MDL) were replaced by half of the MDL values and their uncertainties were set as 5/6 of the MDL values.

## 3. Results and Discussion

### 3.1. Mixing Ratios of VOC Species at Different Sites

Overall, the highest mixing ratios of total VOCs were measured at the ZK point (industrial area), followed by the center city site (ZY). The background point of ML had the lowest ratio. The mixing ratios of the total VOCs in the ZK, ZY, and ML sites were 63.56 and 61.91 ppbv, 52.90 and 39.66 ppbv, and 28.02, and 30.11 ppbv at 9:00 am. and 15:00 pm., respectively. Figure 2 also shows that the high values of alkanes in ZY and ML account for over 35% of the total, followed by oxy-organic VOCs (OVOCs) and alkenes (15–25%).The values of aromatics and halohydrocarbons were less than 15%. By contrast, the highest values of OVOCs (more than 30% of the total VOCs) characterized the ZK (industrial site). The ZK OVOCs likely came from the emissions of automobile spraying and solvent coating given that the site was located in the development zone of Wuhan, which was surrounded by a large number of automobile manufacturing, repair, and other supporting industries, to form a relatively complete automobile and supporting industrial park.

The 10 most abundant VOC species measured at the three sampling sites and other cities of China are listed in Table 2. The results showed that most VOC species concentrations measured at the ZY and ZK sites were slightly higher than those measured at the ML site, except for isoprene. Among the 10 VOC species with the highest concentrations, propane was the most abundant in ZY and ZK, this funding is consistent with that of previous research [21]. The measured concentrations of ethylene, and benzene measured were lower than previous study in Wuhan [19], whereas most of the abundant species were higher than those of a previous study. The propane concentration level-of different cities was significantly higher than that of other cities, especially at the ZY and ZK sites, probably due to vehicular consumption of LPG. The i-pentane concentration level at the ZY and ZK sites was higher than that of Shanghai, Jinan, and Hong Kong, thereby indicating serious motor vehicle emissions in Wuhan in recent years.

### 3.2. Temporal Variationin VOC Speciation

The seasonal variations in VOC species are affected by different emission sources and photochemical and mixing processes. In this section, the seasonal variations in VOC species and five representative VOC ratios are analyzed to obtain the seasonal variation-in the VOC sources and photochemistry shown in Figure 3. The mixing ratio of total VOC was measured in autumn, followed by measurements in summer and winter, this approach was different from many other cities [30,31,32,33]. In addition, the significantly high error range of the mixing ratios the total VOC and five main VOC species in autumn, indicated that the chemical composition of ambient VOCs could be affected by the relative contributions of emission sources and photochemical and mixing processes. Among the five VOC species, alkanes were the dominant species from 8.36 ppbv to 14.67 ppbv during each season throughout the year, and its concentration was higher in autumn and winter while lower in summer. The seasonal variation of alkenes and aromatics were similar to that of alkanes. However, the seasonal variation of oxy-organics was higher in summer but lower in winter. Meanwhile, the concentration difference of halohydrocarbons was weak throughout the year, exhibited the difference-in emission sources in the five VOC groups.

The main source of alkanes is direct emission, of which the increase in the concentration in summer and autumn is mainly due to the significant volatilization-effect under high temperature. Thus, during July and August, the high temperature of Wuhan (exceeding 30 °C) implied high alkane productivity. Combustion in urban regions would emit large amount of olefin, especially in autumn and winter. In 2018, the comprehensive industrial energy consumption of Wuhan reached 24.1904 million tons of standard coal, excluding bulk coal, This finding indicated the important influence of combustion on the chemical composition of VOCs. The concentration of aromatic hydrocarbon was higher in autumn than in the three other seasons, mainly due to the paint and solvent source. No significant seasonal variation existed in halogenated hydrocarbon-because of the influence from environmental temperature and local industrial activity.

The 72 h backward transport trajectory of-airflow over Wuhan at 9:00 a.m. and 15:00 p.m. during the sampling period was generated by using the backward trajectory model HYSPLIT4 (Figure 4). As shown in Figure 4, the major trajectories can be grouped into three directions. Trajectory (1) showed a short transport pattern—starting from Henan, and passing over northern Hubei. Trajectory (2) originated from Xinjiang and showed extremely long transport patterns, across-Shanxi and Henan. Trajectory (3) began in Jiangsu, and passed through Anhui before arriving at Wuhan. In November, the trajectory was mainly from southwest Hubei, which was the shortest distance path of air masses among the months. Overall, the air mass transportation mainly originated in Henan, Hunan and Jiangsu, and showed small-scale and short-distance features. The large-scale and long-distance air transport mainly started from Xinjiang. The short-distance transportation was mainly used in autumn. To certain extent, this finding could explain the significantly higher VOC concentration levels in autumn those that of other seasons (Figure 4).

### 3.3. Correlations between VOC Species

The ratios between the mixing ratios of pairs of ambient VOC species are useful indictors of the major emission sources, photochemical process and the influence of the generating functions. The ratio of ambient mixing ratios of two hydrocarbons with similar chemical reactivity are theoretically equal to those of their relative emission rates from sources [26,34]. Thus, we examined the monthly variations in several groups of VOC species ratio to reveal the typical characteristics of different emissions. Figure 5 shows the monthly variations in the average mixing ratios of isopentane versus acetylene, toluene versus ethene and isoprene versus 1,3-butadiene. The reactivity for these two compounds of hydrocarbon pairs was similar even with different emission sources. Vehicular exhaust is the main source of acetylene, ethene and 1,3-butadiene [35], but gasoline evaporation is also one of the most important contribution sources to isopentane [36]. Meanwhile, toluene is usually applied in shoemaking, furniture, adhesives, printing and other solvent and paint usage [32].Isoprene influenced by biogenic emissions is mainly determined by ambient temperature and solar irradiation [35]. Figure 5a shows that the ratios of isopentane versus acetylene are higher in June and August than that of other months. This finding was highly correlated with averaged ambient temperature given that high temperature could accelerate the evaporation rate of VOCs from paint and gasoline. The variation of toluene versus ethene is similar to the isoprene versus acetylene, which is higher from March to November than that in winter (December to February). The rate of isoprene versus 1,3-butadiene (0.95 ppbv ppbv-1) was lowest in January, more close to the motor vehicle exhaust emission of isoprene/1,3-butadiene ratio (0.3–0.5) measured in Beijing [35], indicating that the motor vehicle exhaust emission was the major emission source of VOCs in January. However, the rate of isoprene versus 1,3-butadiene was in the range from 5 to 35 from May to September, which was over 10 times higher than that in January. This finding demonstrated the clear emission characteristics of the biological source.

### 3.4. VOC Source Identification

#### 3.4.1. Performance of PMF Modeling

Only some VOC species should be-subjected to source identification in PMF. The general principles in choosing the particular species were as follows:(1) The species with signal-to-noise ratio (which indicated whether the variability in measurements was real or within the noise of the data) less than 0.2 must be eliminated (US EPA, 2014). (2) The significant source tracer of species must be included, even those with low concentrations. (3) Highly reactive species should be excluded because they reacted with other short-lived airborne substances, except for source identification species [37,38]. (4) Calculating the source characteristics of species and high concentration species was unnecessary [33]. Therefore, 28 VOC species were selected in PMF to resolve their relative contribution to the VOC concentration of all kinds of emission sources in the atmosphere, including nine kinds of C_2_–C_8_ alkanes, seven types of C_2_–C_5_ olefins, acetylene, five kinds of aromatic hydrocarbons, five types of halogenated hydrocarbons, and methyl *tert*-butyl ether (MTBE).

#### 3.4.2. Pollution Factor Identification—Using the PMF Model

Figure 6 shows the factor (% species) of each pollution source in the entire year. Six factors including solvent usage, gasoline evaporation, industrial process, motor vehicle exhaust emissions, combustion source and biogenic emission, were resolved via PMF analysis.

Factor 1 was closely associated with solvent usage because of high loadings of C_6_–C_7_ aromatic hydrocarbons, especially for-toluene, ethylbenzene, *m/p*-xylene, and *o*-xylene. The factor profiles identified were based on previous studies on source-appointments and VOC source emissions [24,35,38,39].

Factor 2 was identified as gasoline evaporation because of the high fractions of isopentane, with low levels of MTBE. Isopentane is a major constituent of gas oils [40,41]. In this study the average fraction of isopentane in Factor 2 was 43.1%.

The main components of Factor 3 were halogenated hydrocarbon, including methyl chloride, methylene chloride, chloroform, and 1,2-dichloroethane, and 1,2-dichloro propane, the average fractions of which were up to 36.1%, 55.5%, 40.3%, 58.9% and 56.7%, respectively. Such-species were mainly used in daily chemicals and electronic products in the manufacturing industry. Hence, Factor 3 was classified as industrial process.

High loadings of C_2_–C_6_ alkane, alkene (ethylene, propylene, and 1,3-butadiene), acetylene, MTBE, benzene, and toluene were found in Factor 4. MTBE is a kind of high-octane gasoline additive, mainly existing in gasoline products. Isopentane is a tracer in gasoline volatilization, and ethylene, propylene, acetylene and1,3-butadiene are gasoline combustion products [14,42,43]. Therefore, Factor 4 was identified as motor vehicles exhaust. In recent years, vehicle increased by more than 15% per year in Wuhan at 2.6 million in 2016 (Wuhan Statistical Yearbook, 2016). Consequently, vehicle exhaust emission has been a major VOC source.

High loadings of typical combustion emission species in Factor 5, included ethane, propane, acetylene, benzene, and ethylene, with average fractions up to 71.6%, 57.3%, 53.1%, 25.9%, and 25.4%. Therefore, Factor 5 was identified as combustion. Factor 6 was identified as biogenic emission, because isoprene, which is a tracer for biogenic emission, demonstrated had the highest factor loading (40.5%) in the entire year [44,45,46].

#### 3.4.3. Monthly Variation in VOC Sources

Figure 7 shows the average relative contribution of VOC sources in different months. The monthly variation pattern, of vehicle exhaust was unclear with relative contributions ranging from 20% to 32%. The relative contribution of combustion was significantly higher during winter and spring (January to April) at 43–52%likeydue to the significant increase in burning activities, such as straw and civil coal burning in winter. However, the relative contribution of solvent usage, gasoline evaporation, and biogenic emission were large from May to August. Especially in July, the relative contributions of biogenic emission exhibited high values (>20%) because of the high light intensities and ambient temperatures.

#### 3.4.4. Spatial Variation in VOC Sources

Figure 8 illustrates the spatial distributions of relative contributions (%) in VOC sources at the three sites. The extremely high levers of VOCs were combustion, vehicle exhaust, and solvent usage in the ZK site, whereas combustion, vehicle exhaust, and industrial process were the dominant emission sources of ambient air in the ML and ZY sites. Combustion was the largest contributor (>30%) of VOCs in Wuhan. In 2014, the standard coal consumption was 2.307 million tons, which was—higher than those of other cities in China. Combustion, including industrial, civil coal burning, LPG, and biomass burning, was an important VOC source in Wuhan. Meanwhile, the relative contributions of solvent usage exhibited higher values (22.3%) in the ZK site, but lower values (6.5%) in the ML site likely due to the high density of industries in the different sites. As mentioned before, a large number of auto manufacturing and vehicle maintenance companies surrounding the ZK site used a considerable amount of solvents, such as toluene, xylene, and trimethylbenzene, in the car-spraying process. Vehicle exhaust exhibited high (30.6%) contributions in the ZY site likely due to a larger number of vehicles in central Wuhan. The largest contributor (5.0%)of biogenic emissions was found in the ML site (suburban north) with high vegetation coverage. The relative contributions from the two other kinds of sources, including gasoline evaporation, industrial process and combustion, did not show clear spatial distribution characteristics.

## 4. Conclusions

Field measurements of VOCs were conducted in the morning and afternoon at the ZK(residential site), ML (industrial site) and ZY (control site) in Wuhan in central China, from July 2016 to June 2017.We quantified up to 101 VOC species, and the highest mixing ratios of the total VOCs were measured at ZK, followed by ZY, and reasonably the lowest at the background point ML. ZK obtained a very high level of oxy-organics, whereas ZY and ML achieved high mixing ratios of alkanes. The varying composition of VOCs with different seasons reflected the heterogeneous distribution of VOC sources in the region. The monthly variations in VOC ratios (e.g., i-pentane/acetylene, toluene/ethane and isoprene/1,3-butadiene) with similar chemical reactivity indicated the typical characteristics of different emissions. 

The PMF model was applied to identify the possible sources and evaluate the contribution of each emission source to the total VOC concentrations. Combustion was identified as the largest contributor (>30%) of VOCs in different types of points. The relative contributions of solvent usage exhibited higher values (22.3%) in ZK site but lower values (6.5%) in the ML site. Vehicle exhaust exhibited high (30.6%) contributions in the ZY site. The largest contribution (5.0%) of biogenic emission was found in the ML site (suburban north) with high vegetation coverage. The relative contributions from the two other kinds of sources, including gasoline evaporation, industrial process, and combustion, demonstrated unclear characteristics of spatial distribution. Overall, gasoline-related emission (the combination of gasoline exhaust and gas vapor) and combustion were the two most important VOC sources.

## Figures and Tables

**Figure 1 ijerph-17-00791-f001:**
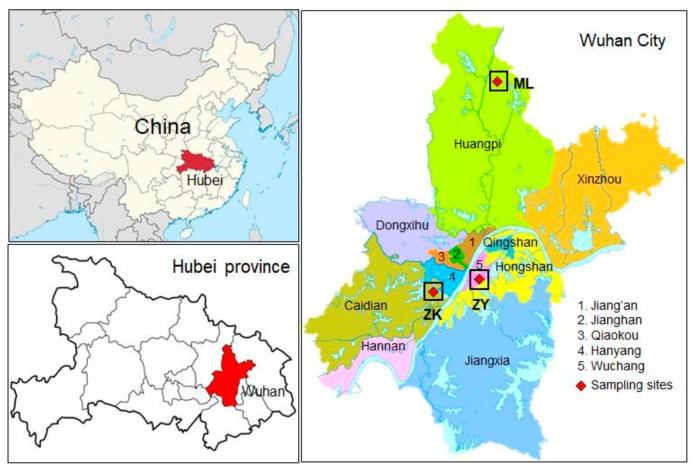
Locations of the sampling sites in Wuhan. ZY: Zi-yang, ZK: Zhuan-kou, ML: Mulan Lake.

**Figure 2 ijerph-17-00791-f002:**
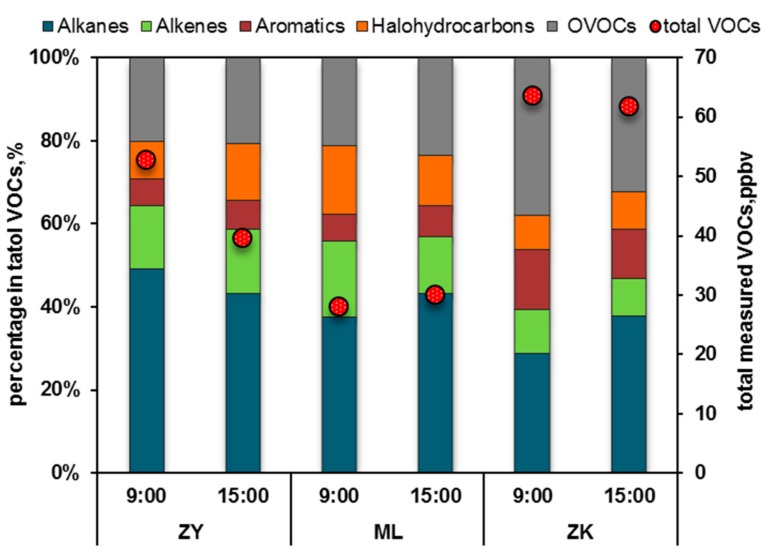
Regional distribution of the mixing ratio and total VOCs measurement at the three sites.

**Figure 3 ijerph-17-00791-f003:**
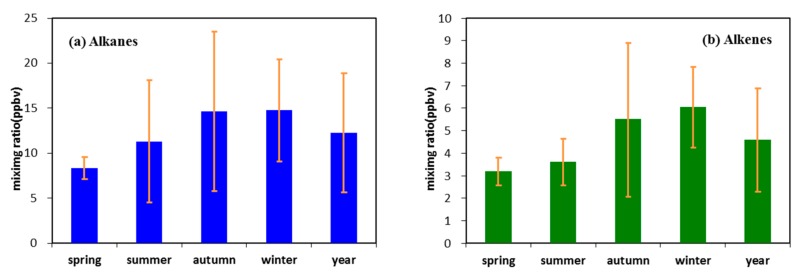
The seasonal variation of alkanes, alkenes, aromatics, halohydrocarbons, OVOCs and TVOCs.

**Figure 4 ijerph-17-00791-f004:**
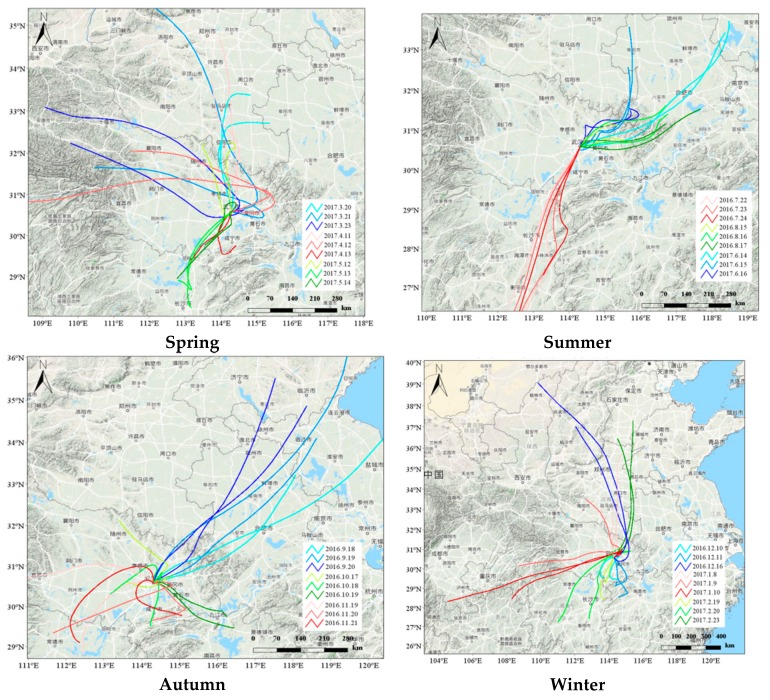
Analytical results of the 72 h air mass back trajectories at 500 m elevation during the monitoring periods, at 9:00 and 15:00 during the sampling period.

**Figure 5 ijerph-17-00791-f005:**
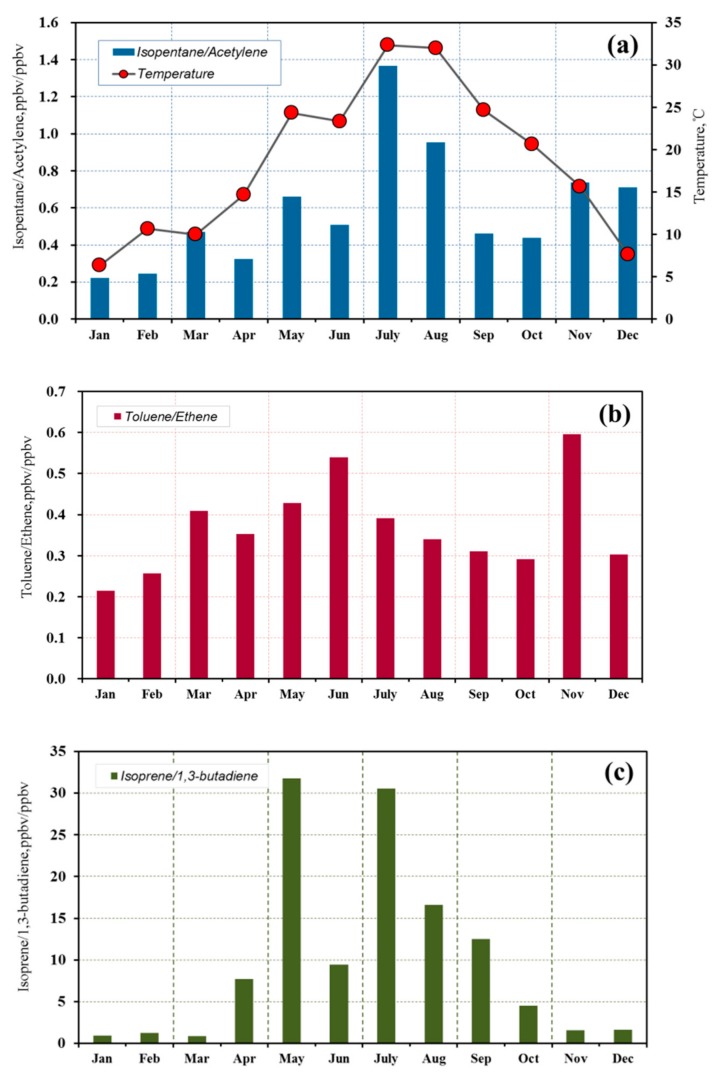
Monthly variations in the average ratios of (**a**) isopentane/acetylene, (**b**) toluene/ethene and (**c**) isoprene/1,3-butadiene with meteorological elements in Wuhan city.

**Figure 6 ijerph-17-00791-f006:**
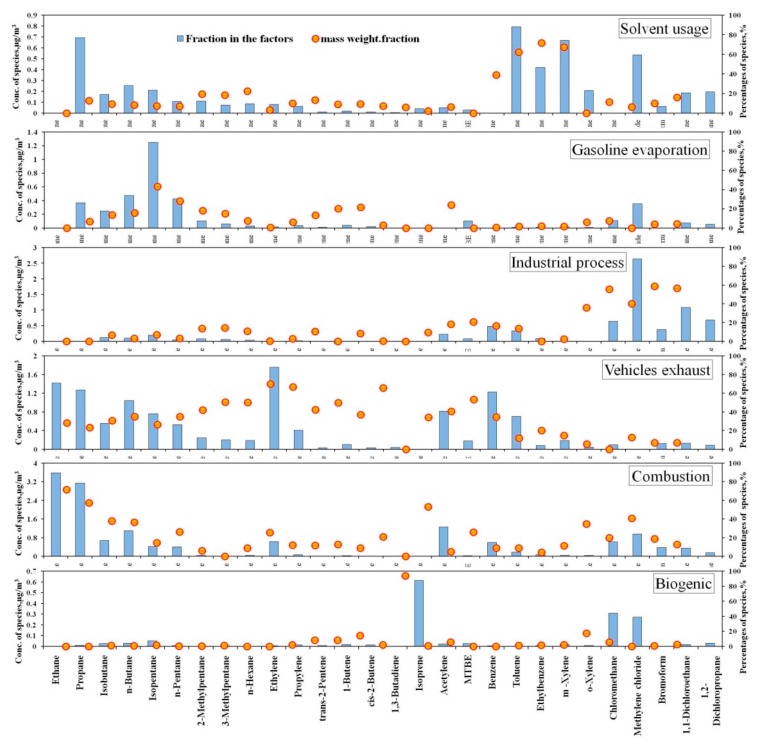
The concentration and factor profiles of each source for the entire year.

**Figure 7 ijerph-17-00791-f007:**
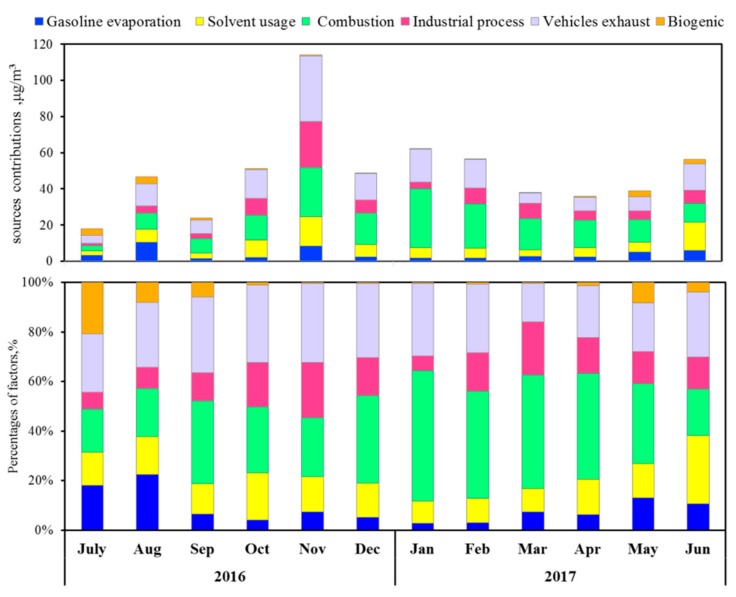
Seasonal variations in the average relative contributions of VOCs sources.

**Figure 8 ijerph-17-00791-f008:**
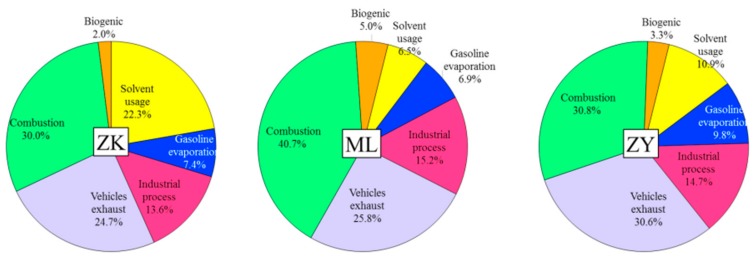
The spatial distributions of the relative contributions (%) in VOCs sources.

**Table 1 ijerph-17-00791-t001:** The meteorological parameters during the sampling periods by month at Wuhan.

Month	Jan	Feb	Mar	Apr	May	Jun	July	Aug	Sep	Oct	Nov	Dec
Temperature (°C)	6.30	10.70	10.00	14.70	24.30	23.30	32.30	32.00	24.70	20.70	15.70	7.70
Relative humidity (%)	78.70	71.30	86.00	79.70	70.70	83.70	65.70	76.00	54.70	79.70	92.30	75.30
Wind speed at 9:00	2.00	3.70	3.30	2.70	3.00	3.00	5.90	3.20	4.50	2.00	1.70	3.30
Wind speed at 15:00	2.70	5.30	4.00	3.30	3.00	2.00	5.90	2.50	3.40	2.00	1.70	4.00

**Table 2 ijerph-17-00791-t002:** The 10 most abundant species (ppbv) measured in Wuhan and other cities.

VOC Species	ZY	ML	ZK	Wuhan ^a^	Beijing ^b^	Shanghai ^c^	Jinan ^d^	Hongkong ^e^
Propane	7.25	2.74	6.91	1.56	3.18	4.56	1.89	2.85
*n*-Butane	2.19	1.20	1.79	0.97	2.46	2.08	1.04	2.69
*i*-Butane	2.16	1.37	1.19	1.27	2.07	1.41	1.72	1.76
*i*-Pentane	2.87	1.42	2.03	0.68	3.68	2.36	1.10	1.37
Ethylene	2.95	1.89	2.47	4.25	4.14	—	1.73	1.08
Propylene	0.68	0.31	0.57	0.10	1.05	0.96	1.88	0.21
Isoprene	0.28	0.23	0.14	0.05	0.37	0.13	0.18	0.78
Acetylene	2.77	1.97	2.48	2.10	4.82	—	—	—
Benzene	0.94	0.67	0.88	2.26	1.56	1.76	0.71	0.52
Toluene	1.14	0.92	3.99	—	2.71	4.62	1.01	2.74
*m/p*-Xylene	0.38	0.09	1.33	0.40	1.80	1.36	0.55	0.60

^a^ [19]; ^b^ [27]; ^c^ [28]; ^d^ [29]; ^e^ [24].

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
