# Peer review of "A Spatial-Temporal Resolved Validation of Source Apportionment by Measurements of Ambient VOCs in Central China"

_ijerph, 2020, doi:10.3390/ijerph17030791_

Round 1

Reviewer 1 Report

Review of Shen et al., A spatial-temporal resolved validation of source apportionment...’

The manuscript by Shen et al. presents observations of the ambient concentrations of a wide variety of VOCs at three locations around Wuhan, a city in central China. In addition to a presentation of the measurements, Positive Matrix Factorization is used to derive dominant source profiles and the seasonal variation in the contribution from these broad source categories is shown.

The analysis of the data and the presentation of the results are relatively straight-forward and I have no serious concerns with the manuscript. In particular, the source profiles identified from the PMF analysis and the seasonal variation of these sources seems quite reasonable. My only one concern about the presentation would be that, particularly in Section 3.2 (lines 163 – 205), there is a significant amount of interpretation derived directly from the absolute concentrations and these concentrations are going to be very significantly affected by the particular meteorology that was present when the measurements were made. The measurements were made on three days each month for one year and from Figure 4 it seems the observations were made on consecutive days each month so that the meteorological conditions of these three days are likely to be similar. As a result, it is very likely the measurements are not a robust sample of the actual month-to-month variations in the concentrations and show some significant effects from random noise. I would suggest the authors add some caveats that the month-to-month variations in concentrations are likely significantly affected by the particular meteorology on the days when the measurements were taken. Of course, the PMF and correlations between species will not be affected by this and the conclusions drawn from this analysis is much stronger.

Minor comments

Lines 21-22: Alkenes appear twice in ‘Alkenes constituted the largest percentage (>35%) in mixing ratios of the quantified VOCs at industrial and residential sites, followed by oxy-organics and alkenes (15%~25%)...’

Lines 70 – 76: I think it would be helpful if you noted how far the Mulan Lake (ML) site is from the central area of Wuhan.

Line 80: ‘July 2016 to June 2017 in the 9:00am and in the 15:00pm.’ could be re-written as ‘July 2016 to June 2017 at 9:00 AM and 15:00 PM local time.’

Line 88: It is a bit odd to start a sentence with a number, so I would suggest ‘...samples. A total of 101 VOCs species...’

Line 122: ‘the amount of mass concentrated by each source to sample’ I think in place of ‘concentrated’ you want ‘contributed’? It would be ‘the amount of mass contributed by each source to the sample’

Lines 175 – 177: The annual variation in the concentration of the halohydrocarbons shown in Figure 3 looks very similar to that of aromatics. How do you justify the statement that ‘the concentration difference of halohydrocarbons was weak through the year’ while not also finding the same for the aromatics?

Line 295 – I cannot find Figure 8 in the PDF file I have.

Author Response

Response to the referee’s comments
Referee A
Comment 1: it is very likely the measurements are not a robust sample of the actual month-to-month variations in the concentrations and show some significant effects from random noise. I would suggest the authors add some caveats that the month-to-month variations in concentrations are likely significantly affected by the particular meteorology on the days when the measurements were taken. Of course, the PMF and correlations between species will not be affected by this and the conclusions drawn from this analysis is much stronger.

Response: Thanks for the referee’s kind suggestion. The specific collection time of each month's environmental samples follows two main principles: On the one hand, the sampling date should try to cover every month in the whole year, including both working days and weekends. The system sampling design is adopted without excessive Human intervention to ensure the time representativeness of the collected samples; on the other hand, sampling in extreme weather such as wind, rainfall and snowfall should be avoided to rule out the accidental effects brought by extreme weather conditions. Of course, there may be a lack of sample representativeness for three days of sampling a month. This work will be improved in subsequent studies.

Comment 2: Alkenes appear twice in ‘Alkenes constituted the largest percentage (>35%) in mixing ratios of the quantified VOCs at industrial and residential sites, followed by oxy-organics and alkenes (15%~25%)...’

Response: Thanks for the referee’s suggestion. I am sorry for writing‘ alkanes ’as ‘alkenes ’due to my mistake, which has been modified in the paper.

Comment 3: I think it would be helpful if you noted how far the Mulan Lake (ML) site is from the central area of Wuhan.

Response: Thanks for the referee’s suggestion. The Mulan Lake (ML) site is about 50Km from the central area of Wuhan, which has been added in the article.

Comment 4: ‘July 2016 to June 2017 in the 9:00am and in the 15:00pm.’ could be re-written as ‘July 2016 to June 2017 at 9:00 AM and 15:00 PM local time.’

Response: Thanks for the referee’s suggestion. It has been modified in the paper.

Comment 5: It is a bit odd to start a sentence with a number, so I would suggest ‘...samples. A total of 101 VOCs species...’

Response: Thanks for the referee’s suggestion. It has been modified in the paper.

Comment 6: ‘the amount of mass concentrated by each source to sample’ I think in place of ‘concentrated’ you want ‘contributed’? It would be ‘the amount of mass contributed by each source to the sample’

Response: Thanks for the referee’s suggestion. It has been modified in the paper.

Comment 7: the annual variation in the concentration of the halohydrocarbons shown in Figure 3 looks very similar to that of aromatics. How do you justify the statement that ‘the concentration difference of halohydrocarbons was weak through the year’ while not also finding the same for the aromatics?

Response: Thanks for the referee’s suggestion. I'm very sorry for some measures taken due to my negligent halogenated hydrocarbon concentration. I have changed the correct concentration in the paper. It can be seen that the seasonal variation characteristics of halogenated hydrocarbons are not obvious. Compared with halogenated hydrocarbons, the seasonal variation characteristics of aromatic hydrocarbons are relatively significant.

Comment 8: I cannot find Figure 8 in the PDF file I have.

Response: Thanks for the referee’s suggestion. The Figure 8 has been added in the article.

Reviewer 2 Report

 In this paper, the authors conducted offline VOCs measurements at residential, industrial and background sites in Wuhan city from July 2016 to June 2017. They demonstrated that gasoline-related emission was the most important VOCs source in industrial and residential area, with the relative contribution of 32.1% and 40.4%, reactively. I would like to give a recommendation to publish this manuscript.  Howevere, There are a few grammar and spelling mistakes, such as Page 2, line 50 to line 53. It is required that substantial revisions should be done before publish.

Author Response

Response to the referee’s comments
Referee B
Comment 1: There are a few grammar and spelling mistakes, such as Page 2, line 50 to line 53. It is required that substantial revisions should be done before publish.

Response: Thanks for the referee’s suggestion. I have made meticulous changes to the large number of grammatical and spelling errors you raised, and thank you again for your suggestions.